# Efficient Cross-Task Prompt Tuning for Few-Shot Conversational Emotion Recognition

**Yige Xu**[1,2], **Zhiwei Zeng**[1,2], **Zhiqi Shen**[2]

[1]Joint NTU-UBC Research Centre of Excellence in Active Living for the Elderly
[2]School of Computer Science and Engineering
Nanyang Technological University, Singapore
yige002@e.ntu.edu.sg, {zhiwei.zeng,zqshen}@ntu.edu.sg

## Abstract

Emotion Recognition in Conversation (ERC) has been widely studied due to its importance in developing emotion-aware empathetic machines. The rise of pre-trained language models (PLMs) has further pushed the limit of ERC performance. However, most recent works on ERC using PLMs are heavily data-driven and require fine-tuning the entire PLMs. To improve both sample and computational efficiency, we propose a derivative-free optimization method called **C**ross-**T**ask **P**rompt **T**uning (CTPT) for few-shot conversational emotion recognition. Unlike existing methods that learn independent knowledge from individual tasks, CTPT leverages sharable cross-task knowledge by exploiting external knowledge from other source tasks to improve learning performance under the few-shot setting. Moreover, CTPT only needs to optimize a vector under the low intrinsic dimensionality without gradient, which is highly training-efficient compared with existing approaches. Experiments on five different contextual conversation datasets demonstrate that our CTPT method has superior results on both few-shot scenarios and zero-shot transfers.

## 1 Introduction

Emotion Recognition in Conversation (ERC) detects emotion categories (e.g., *netural*, *happiness*, *sadness*) of each utterance in a given textual conversation. As pre-trained language models (PLMs) (Devlin et al., 2019; Liu et al., 2019) have brought a huge breakthrough to natural language processing (NLP), PLMs are also increasingly employed by ERC models as encoders to improve recognition performance (Zhong et al., 2019; Kim and Vossen, 2021; Chudasama et al., 2022). However, the performance gain from PLMs is often achieved at the exorbitant cost of expensive training and fine-tuning processes. PLM-based ERC models tend to suffer from poor sample efficiency

and computational efficiency as they often involve a large number of training examples and millions of trainable parameters, which potentially prevents current PLM-based models from achieving their best performance in low-resource scenarios.

Few-shot learning techniques (Motiian et al., 2017; Wang et al., 2021) hold the promise to improve both sample and computation efficiency for deploying PLMs in new scenarios where data can be limited. Recently, prompt tuning (Li and Liang, 2021; Lester et al., 2021), which trains a set of discrete or continuous prompt embeddings conditioned on a frozen PLM, has shown promising results in few-shot learning settings (Gao et al., 2021; Gu et al., 2022; Guo et al., 2022). The prompt can be regarded as a way to retrieve the knowledge already memorized in the PLM. The effectiveness of prompts lies in their capability to adapt to new tasks while preserving the knowledge embedded in PLMs, without causing overfitting issues that can arise from full-model fine-tuning (Liu et al., 2021).

However, most recent works on ERC are large-scale data-driven that focus on the full dataset setting (Lee and Choi, 2021; Song et al., 2022a). Guibon et al. (2022) firstly explore the few-shot ERC task, but their setting is not strictly few-shot, which may lead to a variety of examples for each label. For example, their training set contains more than $k$ examples for each label under the $k$-shot setting.

To this end, we strictly define the ERC task under the few-shot setting and propose a Cross-Task Prompt Tuning (CTPT) solution. Existing prompt tuning methods independently learn task-specific knowledge from each task, yet such knowledge is often very limited in the few-shot setting. Our proposed CTPT leverages cross-task knowledge by exploiting external knowledge from other source tasks to improve learning performance under the few-shot setting. The cross-task knowledge from other source tasks can be divided into two parts: external task-specific knowledge and emotional

knowledge. For external task-specific knowledge, we utilize a multi-head attention module (Vaswani et al., 2017) to learn knowledge from source tasks. For emotional knowledge, we combine the same emotion within different textual categories from different tasks and then reformulate the verbalizer that decodes the output to the label distribution.

One limitation of prompt tuning is that it involves backpropagating the loss through all the Transformer layers of a PLM for every batch even though we freeze the PLM, which can lead to computational inefficiency. To further improve the computational efficiency of PLM-based ERC models, we optimize a vector with intrinsic dimensionality (Li et al., 2018) instead of the whole continuous prompt, which reduces the number of parameters from hundreds of thousands to about 1,000. Following Sun et al. (2022), we use a Covariance Matrix Adaptation Evolution Strategy (CMA-ES) (Hansen and Ostermeier, 2001; Hansen et al., 2003) to optimize the parameters, which is derivative-free. With the derivative-free optimization, we separate our approach from the PLM and do not require backpropagation for parameter learning.

Compared with single-task prompt tuning, our proposed CTPT method can utilize external knowledge from other tasks to boost the performance of the target task. Experiments under the few-shot scenarios and the zero-shot transfer show that CTPT can obtain a better result. In addition to this, our proposed CTPT is derivative-free, which does not need backpropagation. Compared with derivative-based backpropagation, the experiment result shows that CTPT can obtain comparable results without derivative information.

The main contributions of this paper are summarized as follows:

(1) To the best of our knowledge, we are the first to strictly define and tackle the few-shot setting for the ERC task. We propose a Cross-Task Prompt Tuning (CTPT) method that can efficiently learn and utilize cross-task knowledge.

(2) To improve the training efficiency, we use the derivative-free optimization algorithm to optimize the parameter. It skips the backpropagation stage and does not require gradient information.

(3) Our proposed CTPT only needs to optimize about 1,000 parameters, which is much more training-efficient than any other existing PLM-based ERC method.

(4) Our proposed CTPT is trained under the few-shot setting, which is sample-efficient. CTPT can also obtain a better experimental result on zero-shot transfer, which can be deployed in new scenarios with limited training examples.

## 2 Related Works

### 2.1 Emotion Recognition in Conversation

Early studies on ERC mainly utilized audio-based features (Lee and Narayanan, 2005) or lexicon-based features (Devillers and Vidrascu, 2006). Recently, there are a series of deep learning approaches focused on emotion recognition in conversational videos or multi-turn Tweets (Hazarika et al., 2018; Zahiri and Choi, 2018; Zhong et al., 2019; Ishiwatari et al., 2020). In recent years, PLM has been increasingly applied in ERC models (Lee and Choi, 2021; Shen et al., 2021; Song et al., 2022a). A commonality among these prior approaches is their shared approach on the integration of various forms of external knowledge to enhance emotion detection, including knowledge from knowledge base (Zhong et al., 2019), knowledge from commonsense (Ghosal et al., 2020; Yi et al., 2022), knowledge from multi-modal (Li et al., 2022), and inherent knowledge within PLM (Kim and Vossen, 2021). Unlike existing methods that focus on enriching task-specific knowledge only, we also explore sharable cross-task knowledge from other source tasks.

### 2.2 Prompt Tuning

Despite the success of GPT-3 (Brown et al., 2020) with 175 billion parameters, it has become increasingly difficult and expensive to utilize such big language models. One possible solution to leverage large pre-trained models is parameter-efficient tuning methods, such as prompt-tuning (Lester et al., 2021; Li and Liang, 2021). In prompt tuning, downstream tasks are reformulated as a language modelling task with the help of a textual prompt. For example, a classification task that aims to predict the emotion category of a given sentence can be reformulated as: "I felt so [MASK], [X]". Here [X] is the given sentence, [MASK] is the mask token that PLM needs to predict, and "I felt so [MASK]" is the template of prompting. The aforementioned prompt consists of discrete tokens, which are also known as a hard prompt. There is another prompt named soft prompt (Qin and Eisner, 2021), which consists of continuous embeddings. Recently, prompt tuning has been proven successful in both few-shot

scenarios (Gu et al., 2022; Vu et al., 2022) and zero-shot transfer (Guo et al., 2022).

Although prompt tuning has brought success in many NLP domains such as text classification (Gao et al., 2021), question answering (Yang et al., 2022), and commonsense reasoning (Liu et al., 2022), Yi et al. (2022) first practising prompt tuning on the ERC task that utilizes learnable continuous prompt to model the relationship between contextual information and commonsense knowledge. In this paper, we utilize learnable prompts to model the relationship between emotional categories among different tasks under the few-shot setting.

### 2.3 Derivative-Free Optimization

Different from many neural networks that require gradient information for backpropagation, derivative-free optimization (DFO) algorithms aim to obtain optimal solutions without derivative information. Most DFO algorithms (Hansen et al., 2003; Shahriari et al., 2016) are under the sampling-and-updating structure, which firstly samples a solution $x$ and then optimize the parameters via the function values $f(x)$. In recent years, DFO algorithms have been applied to many downstream areas, such as automatic machine learning (Snoek et al., 2012), and reinforcement learning (Salimans et al., 2017). More recently, Sun et al. (2022) proposed a DFO method to optimize continuous prompts without gradient information. In this paper, we further extend the DFO method to optimize not only the continuous prompt but also the parameters for cross-task learning.

## 3 Methodology

### 3.1 Problem Definition and Notations

In this section, we will briefly define the emotion recognition in conversation (ERC) task and the ERC task under the few-shot setting.

The full dataset setting contains the conversation set $\mathcal{X} = \{\mathbf{x}_1, \mathbf{x}_2, \cdots, \mathbf{x}_n\}$ with $n$ different conversations as well as the emotion category set $\mathcal{Y} = \{\mathbf{y}_1, \mathbf{y}_2, \cdots, \mathbf{y}_n\}$. The target is to predict the corresponding emotion category set $\mathcal{E} = \{\mathbf{e}_1, \mathbf{e}_2, \cdots, \mathbf{e}_n\}$.

More specifically, the input of the task is a conversational content $\mathbf{x}_i = [x_1^i, x_2^i, \cdots, x_{|\mathbf{x}_i|}^i]$. The output of the task is an emotional category set $\mathbf{e}_i = [e_1^i, e_2^i, \cdots, e_{|\mathbf{x}_i|}^i]$. The ground-truth is also an emotional category set $\mathbf{y}_i = [y_1^i, y_2^i, \cdots, y_{|\mathbf{x}_i|}^i]$. The target of the task is to predict the emotional

category for each utterance to maximum match the ground-truth emotion category. In the $i$-th conversation, $x_j^i = [x_{j,1}^i, x_{j,2}^i, \cdots, x_{j,|x_j^i|}^i]$ indicates the $j$-th utterance, $x_{j,k}^i$ indicates the $k$-th token in the $j$-th utterance, $|x_j^i|$ indicates the sequence length of the $j$-th utterance, and $|\mathbf{x}_i|$ is the number of utterances in the $i$-th conversation.

Dataset under the few-shot setting is a subset that under the full dataset setting. The new dataset under the $k$-shot setting is marked as $\{(x, y)|x \in \hat{\mathcal{X}}, y \in \hat{\mathcal{Y}}\}_k$, where $\hat{\mathcal{X}}$ and $\hat{\mathcal{Y}}$ indicate the input sequence as well as the emotion category for the new training set. Correspondingly, the predicted emotion category is $\hat{\mathcal{E}}$. Here $k$-shot indicates that there are $k$ training examples for each emotion category. We randomly select the new dataset and keep it the same in the following experiments. Similar to the full dataset setting, the aims under the few-shot setting is to predict the emotion category $\hat{\mathbf{e}}_i$.

Under the few-shot setting, the training set as well as the development set, are sampled randomly from the vanilla dataset of the full dataset setting, while the testing set keeps unchanged. At the beginning of the training stage, we first randomly select some training examples under the following rules: for each given emotion category (e.g., *netural*, *happiness*, *sadness*, etc.), we randomly select $k$ utterances from the vanilla training set. In other words, we keep the textual conversational content but retain only one emotion category for one conversation. Therefore, for each training example, the input content remains the conversation content $\hat{\mathbf{x}}_i = [x_1^i, x_2^i, \cdots, x_{|\mathbf{x}_i|}^i]$, and the ground-truth becomes $\hat{\mathbf{y}}_i = y_j^i$ that $j$ is randomly selected before the training stage.

All experiments are conducted on few-shot settings ($k = 16$). For each dataset, we sample the subset of the training set and the development set and keep the testing set unchanged. For a fair comparison, all baselines and CTPT are trained by the same training set.

### 3.2 Overview of the Model

In this section, we will briefly introduce the overview of the whole model. The input of CTPT is a textual sequence that contains the conversational context, and the output of CTPT is an emotion label. CTPT can be mainly divided into three parts: task-specific prompt tuning (TSPT), cross-task prompt learning (CTPL), and cross-task prompt observation (CTPO). The overall architecture is shown in

Figure 1: Overall architecture of our proposed CTPT model.

Figure 1.

The first part of CTPT is TSPT. In this part, we learn task-specific knowledge from different source tasks [1], which is harnessed later to the target task [2]. Then, we have CTPL that employs an attention module to learn the external task-specific knowledge learned by TSPT from source tasks and the emotional knowledge from commonsense. Lastly, we have CTPO that utilizes a gate-like mechanism to summarize pertinent cross-task knowledge learned by CTPL. In summary, we have "TSPT + CTPL + CTPO = CTPT".

We concatenate the prompt with summarized cross-task knowledge $\hat{\mathbf{p}}^i$ as well as the input sequence $x$, and then pass it into the PLM. After we obtain the logits of the [MASK] token from PLM, we first decode the logits to word distribution, then map the word distribution to the emotion label, which is the output of the whole model.

In the derivative-free optimization, learnable parameters are contained a vector $\mathbf{z}$ with intrinsic dimensionality. The parameters in the neural network are computed by a linear projection from the vector $\mathbf{z}$. We use the cross-entropy function to compute the loss between logits and the ground-truth label and then use Covariance Matrix Adaptation Evolution Strategy (CMA-ES) (Hansen and Ostermeier, 2001; Hansen et al., 2003) to optimize $\mathbf{z}$.

### 3.3 Task-Specific Prompt Tuning

To address the computational efficiency problem, prompt tuning (Li and Liang, 2021; Lester et al., 2021) is a promising solution. Before CTPT, we use prompt tuning methods to learn knowledge for

the target task, named task-specific prompt tuning (TSPT). Similar to the existing prompt tuning methods, we use soft prompt (Qin and Eisner, 2021) as the template, which can be formulated as:

$$p(y|x) = v(\text{PLM}(P(x))), \quad (1)$$
$$P(x) = \text{concat}[\mathbf{p}; x], \quad (2)$$

where $\text{concat}[\cdot; \cdot]$ is the concatenation, $\text{PLM}(\cdot)$ indicates a pre-trained language model, $P(\cdot)$ is the pattern projective function that converts the input sequence $x$ into a phrased sequence with a [MASK] token. Here $v(\cdot)$ is the verbalizer injective function that decodes the label by the predicted distribution of the [MASK] token, which can be formulated as:

$$v(\mathbf{h}) = g\Big(p([\text{MASK}] = v|\mathbf{h})|v \in \mathcal{V}_i\Big), \quad (3)$$

where $\mathbf{h} = \text{PLM}(P(x))$ is the hidden states outputed from a PLM, $\mathcal{V}_i$ is the verbalizer set for target task $i$, and $g(\cdot)$ is a function trasnsforming the probability of $v$ to the probability of the label. Here different task has different verbalizer.

The soft prompt $\mathbf{p} \in \mathcal{R}^{n \times d}$ in Eq (2) is a learnable matrix, and the objective function is:

$$\mathbf{p}^\star = \underset{\mathbf{p} \in \mathcal{R}^{n \times d}}{\arg\min} \mathcal{L}(\hat{\mathcal{Y}}, \hat{\mathcal{E}}), \quad (4)$$

where $\mathcal{L}$ is the cross-entropy loss function, $\hat{\mathcal{Y}}$ and $\hat{\mathcal{E}}$ are defined in Section 3.1, and $n$ is the number of prompt tokens. The soft prompt can be regarded as a task-specific embedding that contains latent knowledge from the specific task.

### 3.4 Cross-Task Prompt Learning

With TSPT, we obtain a task-specific prompt $\mathbf{p}_t^i$ for the $i$-th task, which contains the task-specific

---

[1] Source tasks indicate tasks exclude the target task $i$.
[2] Target task indicates the task $i$ for evaluation.

knowledge of the target task. The independently learned task-specific knowledge is usually limited under the few-shot setting. One promising solution to address this problem is to introduce abundant sharable knowledge from other source tasks. The sharable knowledge includes external task-specific knowledge from source tasks and prior emotional knowledge learned from commonsense.

**External Task-Specific Knowledge**  Since task-specific knowledge is often very limited in the few-shot setting, we introduce external task-specific knowledge from other source tasks. As aforementioned, the external task-specific knowledge is stored in the prompt learned by TSPT. Therefore, we modify the Equation (2) for the $i$-th task as follows:

$$P(x) = \mathrm{concat}[f(\mathbf{p}_c^i, \mathbf{p}_t^i); x], \qquad (5)$$

where $\mathbf{p}_c^i$ indicates the cross-task prompt for the $i$-th task, and $f(\cdot)$ indicates the combination of task-specific prompt and cross-task prompt.

Inspired by the success of the attention mechanism, we utilize a multi-head attention module to decide what kind of knowledge should be collected from the source tasks. In multi-head attention, the query term is the task-specific prompt from the target task. The key term, as well as the value term, are the task-specific prompt from each source task. The whole module is formulated as:

$$\mathbf{p}_c^i = \sum_{j, j \neq i} \mathrm{MHA}(\mathbf{p}_t^i, \mathbf{p}_t^j), \qquad (6)$$

$$\mathrm{MHA}(\mathbf{p}_t^i, \mathbf{p}_t^j) = \sum_{\mathrm{head}} \mathrm{softmax}\Big(\frac{QK^T}{\sqrt{d}}\Big)V,$$

where $\mathrm{MHA}(\cdot, \cdot)$ indicates the multi-head attention, $d$ indicates the dimension of hidden state. Here $Q$, $K$, $V$ are:

$$\begin{aligned} Q &= W^Q \mathbf{p}_t^i, \qquad\qquad (7) \\ K &= W^K \mathbf{p}_t^j, \\ V &= W^V \mathbf{p}_t^j. \end{aligned}$$

In this module, $W^Q$, $W^K$, and $W^V$ are learnable parameters projected by a learnable vector $\mathbf{z}$ (More details about optimization are shown in Section 3.6).

**Emotional Knowledge**  In order to facilitate the ERC task, we also introduce emotional knowledge

collected from commonsense in addition to external task-specific knowledge. Across different ERC datasets, varying labels might be used to denote identical emotional states. For example, DailyDialog uses "happiness" while MELD uses "joy" to represent the state of being happy. Given this understanding, we can learn cross-task emotional knowledge from disparate tasks encompassing the same emotional state, notwithstanding the divergence in emotion labels, by modifying the verbalizer. Therefore, Eq (3) can be modified as:

$$\begin{aligned} \hat{\mathcal{V}} &= \{v | \forall v \in \mathcal{V}_i, i = 1, 2, \cdots, n\}, \quad (8) \\ h &: \hat{\mathcal{V}} \to \mathcal{V}, \\ v(\mathbf{h}) &= g\Big(p([\mathtt{MASK}] = v | \mathbf{h}) | v \in \mathcal{V}\Big), \end{aligned}$$

where $h$ is a mapping function that maps $v$ from different task-specific verbalizers to a union verbalizer, and $n$ is the number of tasks. With the new verbalizer, the model can learn knowledge from source tasks under the same emotion.

### 3.5 Cross-Task Prompt Observation

In cross-task prompt learning, we obtain a prompt $\mathbf{p}_c^i$ that contains the external task-specific knowledge collected from other source tasks and emotional knowledge collected from commonsense. Similarly to us, Asai et al. (2022) also utilizes an input-attention module to combine multiple source prompts. However, Asai et al. (2022) only considers how to learn prompts from source tasks. We empirically notice that part of the learned knowledge is beneficial to the target task while the other part is useless. To address this problem, we propose an extra stage: cross-task prompt observation.

In the cross-task prompt observation stage, more knowledge from the source task will be observed if it is helpful to improve the validation performance of the target task, while less in contrast. Formulatedly, we optimize a vector $\mathbf{g}$ as a gate-like controller via the derivative-free optimization mentioned in Section 3.6. Thus, the final prompt becomes:

$$\hat{\mathbf{p}}^i = f(\mathbf{p}_c^i, \mathbf{p}_t^i) = \mathbf{g}_i \otimes \mathbf{p}_t^i + (\mathbb{I} - \mathbf{g}_i) \otimes \mathbf{p}_c^i, \quad (9)$$

where $\otimes$ indicates the token-level element-wise multiple, $\mathbb{I}$ is an all one vector, and $\mathbf{g}_i$ are learnable parameters for $i$-th task learned by the following objective function:

$$\mathbf{g}_i^\star = \underset{\mathbf{g}_i \in \mathcal{Z}}{\arg\min} \, \mathcal{L}(\{p(y|x) | x \in \hat{\mathcal{X}}\}, \hat{\mathcal{Y}}). \qquad (10)$$

## 3.6 Derivative-Free Optimization

According to Li et al. (2018), the intrinsic dimensionality is the minimum number of parameters needed to obtain comparable results. Sun et al. (2022) also shows the efficiency of derivative-free optimization for intrinsic dimensionality vector in prompt tuning. To improve the computational efficiency, instead of the derivative-based backpropagation, we utilize a Covariance Matrix Adaptation Evolution Strategy (CMA-ES) (Hansen and Ostermeier, 2001; Hansen et al., 2003) to optimize a vector with intrinsic dimensionality. In each optimization step, the optimizer will first sample some solutions of the learnable vector $\mathbf{z}$. Then we can calculate the loss of each solution that is used by the optimizer to suggest a new $\mathbf{z}$.

To adapt our proposed CTPT, we modify the optimization step as follows:

**Task-Specific Prompt Tuning** In this step, we follow Sun et al. (2022) to compute the task-specific prompt $\mathbf{p}_t$ by a learnable vector $\mathbf{z}$:

$$\mathbf{p}_t = \mathbf{A}\mathbf{z} + \mathbf{p}_0, \tag{11}$$

where $\mathbf{A}$ is randomly initialized and fixed, and $\mathbf{p}_0$ is the initialized prompts from most widely-used tokens.

**Cross-Task Prompt Learning** In this step, we optimize a vector $\mathbf{z}'$ instead of the parameters in the multi-head attention module. Similar to the optimization of TSPT, we firstly project $\mathbf{z}'$ to the parameters space and then separate the parameter space by:

$$\mathbf{W} = \mathrm{concat}[\hat{W}^Q, \hat{W}^K, \hat{W}^V], \tag{12}$$

where $\mathbf{W} = \mathbf{A}'\mathbf{z}'$ that $\mathbf{A}'$ is also randomly initialized and fixed. Then, we reshape the parameters and add a randomly initialized and fixed term:

$$W^Q = \hat{W}^Q + W_0^Q, \tag{13}$$
$$W^K = \hat{W}^K + W_0^K,$$
$$W^V = \hat{W}^V + W_0^V.$$

**Cross-Task Prompt Observation** In this step, we optimize the vector $\mathbf{z}''$ in the same way as the vector $\mathbf{z}$ being optimized in TSPT:

$$\mathbf{g} = \mathbf{A}''\mathbf{z}'' + \mathbf{g}_0, \tag{14}$$

where $\mathbf{A}''$ and $\mathbf{g}_0$ are fixed.

## 4 Experiments

### 4.1 Datasets

We conduct experiments on five widely-used public datasets to show the efficiency of CTPT, including: **EC** (Chatterjee et al., 2019), **DailyDialog** (Li et al., 2017), **MELD** (Poria et al., 2019), **EmoryNLP** (Zahiri and Choi, 2018), and **IEMOCAP** (Busso et al., 2008). Detailed statistics are shown in Table 1. Though some of the datasets are multi-modality, we only utilize the textual information as the input for a fair comparison with baselines.

### 4.2 Baselines

For a comprehensive performance evaluation, we select the following four baselines for comparison:

**KET** (Zhong et al., 2019) The KET is a knowledge-enriched transformer model specifically designed for Emotion Recognition in Conversation (ERC). It employs a knowledge base to infuse external knowledge, which is a representative baseline model before the PLM decade.

**TUCORE-GCN** (Lee and Choi, 2021) The TUCORE-GCN model is a turn-context aware graph convolutional network designed for ERC. It incorporates both a PLM encoder and a graph convolutional network, making it an exemplary representation of PLM-based baseline models.

**EmotionFlow** (Song et al., 2022b) The EmotionFlow is a PLM-based model with an additional CRF layer to capture the emotion transition probability among different utterances.

**SPCL** (Song et al., 2022a) The SPCL is a PLM-based model using supervised prototypical contrastive learning loss, focusing primarily on imbalanced classification problems. It has achieved state-of-the-art results on MELD and EmoryNLP.

### 4.3 Implementation Details

In this paper, we use a soft prompt extended to the input and freeze the PLM. We use CMA-ES (Hansen and Ostermeier, 2001; Hansen et al., 2003) algorithm to optimize the parameters. We choose T5 (Raffel et al., 2020) as our backbone model. All few-shot settings share the same training and development set with $k = 16$ following the settings of few-shot prompt tuning (Gao et al., 2021; Sun et al., 2022).

| Dataset | Domain | # Emotions | # Conv. | # Utter. |
|---|---|---|---|---|
| EC (Chatterjee et al., 2019) | Tweet | 4 | 30,160/2,755/5,509 | 90,480/8,265/16,527 |
| DailyDialog (Li et al., 2017) | Daily Chat | 7 | 11,118/1,000/1,000 | 87,170/8,069/7,740 |
| MELD (Poria et al., 2019) | TV Show Scripts | 7 | 1,038/114/280 | 9,989/1,109/2,610 |
| EmoryNLP (Zahiri and Choi, 2018) | TV Show Scripts | 7 | 659/89/79 | 7,551/954/984 |
| IEMOCAP (Busso et al., 2008) | Daily Chat | 6 | 100/20/31 | 4,758/1,000/1,622 |

Table 1: Statistics of five ERC datasets. $a/b/c$ indicates the number of examples in the training set, development set, and testing set, respectively.

Following Sun et al. (2022) and Lester et al. (2021), we use the soft prompt template and only train the continuous prompts extended to the input texts while freezing the PLM parameters. We utilize a Covariance Matrix Adaptation Evolution Strategy (CMA-ES) (Hansen and Ostermeier, 2001; Hansen et al., 2003) to optimize the parameters without gradient information.

We use the soft template extended to the input sequence with a length of 50. The last token of the template is set as <unk> so that the model can better predict the masked token. Since the task is reformulated as a generalization task with the text-to-text format, we choose T5 (Raffel et al., 2020) as our backbone model.

### 4.4 Evaluation Metric

For EC and DailyDialog, due to the imbalance distribution of categories (more than 80% examples are neutral emotion), we use micro-averaged $F_1$ score excluding neutral category following Chatterjee et al. (2019). For the rest three datasets, following Majumder et al. (2019), we use weighted macro-$F_1$ score. The overall evaluation setting is the same as Zhong et al. (2019).

## 5 Results and Analysis

### 5.1 Main Results

We compare the performance of CTPT against the baselines aforementioned in Section 4.2. We first re-implement the baseline models and achieve the similar performance reported in the original paper. Then we modify the preprocessing code to let all baselines be trained under the same few-shot setting. The result under the few-shot setting is shown in Table 2.

Compared with the baseline models, task-specific prompt tuning (TSPT) outperforms fine-tuning PLMs on most tasks under the few-shot setting. Meanwhile, benefiting from the cross-task knowledge, our proposed cross-task prompt tuning (CTPT) obtains an improvement compared

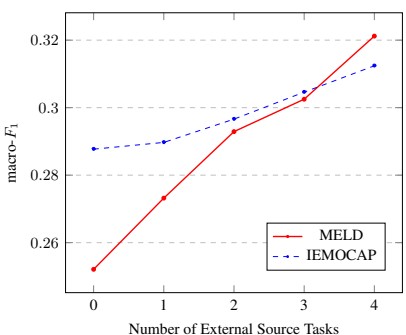

Figure 2: Impacts of removing external source tasks for MELD and IEMOCAP.

with TSPT. Specifically, in addition to EC that TSPT has already obtained a high performance under the few-shot setting, CTPT brings significant improvement compared with TSPT, which demonstrates the effectiveness of utilizing cross-task knowledge. Since DailyDialog and MELD share the same emotion labels, CTPT obtains the most gain on these two datasets, which shows that our cross-task prompt tuning can learn emotional knowledge from the labels.

### 5.2 Model Analysis

To gain deeper insights into CTPT from diverse angles, we undertake analytical experiments in a few-shot setting, where $k = 16$, in this section.

**Analysis in Training Stage** Since the training stage of CTPT is different from other approaches, it is worthwhile exploring the training stage.

First, to prevent the validation performance degradation brought by DFO algorithms, we train CTPT with backpropagation methods. As shown in Table 2, CTPT optimized by DFO algorithms (CTPT w/o. BP) has a comparable result with that optimized by backpropagation methods (CTPT w. BP). In some tasks such as MELD, DFO algorithms perform better than backpropagation methods. The experiment result shows that CTPT obtains comparable results without derivative information, which can be deployed in non-GPU devices.

| | Model | EC | DailyDialog | MELD | EmoryNLP | IEMOCAP |
|---|---|---|---|---|---|---|
| Baselines | KET (Zhong et al., 2019) | 0.1296 | 0.0909 | 0.0897 | 0.1312 | 0.1646 |
| | TUCORE-GCN (Lee and Choi, 2021) | 0.1918 | 0.2029 | 0.2596 | 0.1311 | 0.1527 |
| | EmotionFlow (Song et al., 2022b) | 0.4084 | 0.3749 | 0.2934 | 0.1465 | 0.1699 |
| | SPCL (Song et al., 2022a) | 0.4269 | 0.3699 | 0.2941 | 0.1499 | 0.1873 |
| | TSPT | 0.6274 | 0.4996 | 0.2521 | 0.1613 | 0.2877 |
| Ours | TSPT + CTPL | 0.6226 | 0.5193 | 0.2732 | 0.1724 | 0.2829 |
| | CTPT (w/o. BP) | 0.6394 | 0.5571 | **0.3212** | 0.1902 | 0.3124 |
| | CTPT (w. BP) | **0.6405** | **0.5588** | 0.3128 | **0.2057** | **0.3182** |

Table 2: Performance of different ERC datasets under the few-shot settings ($k = 16$). "TSPT" indicates task-specific prompt tuning, "CTPT" indicates cross-task prompt tuning. The result of EC and DailyDialog are micro-averaged $F_1$, and the result of other datasets are weighted macro-$F_1$. We **bolded** the best result and underline the second best.

| | Micro-$F_1$ | Training Time | GPU Memory Usage |
|---|---|---|---|
| KET | 0.0909 | 4.5 mins | 1.2 GB |
| EmotionFlow | 0.3749 | 7.5 mins | 8.7 GB |
| SPCL | 0.3699 | 7 mins | 7.2 GB |
| CTPT | 0.5571 | 6 mins | 2.8 GB |

Table 3: Comparison of resources requirements on DailyDialog.

| | DailyDialog | IEMOCAP |
|---|---|---|
| TSPT | 0.4996 | 0.2877 |
| CTPT | | |
| w/o. EK | 0.5481 | 0.3076 |
| w. EK | 0.5571 | 0.3124 |

Table 4: Ablations of emotional knowledge. Here "EK" indicates "Emotional Knowledge".

Second, to explore the training efficiency of CTPT, we compare CTPT and other baseline models in terms of training resource requirements. Simply, we compare the two main factors: training time and GPU memory usage. All the methods are implemented with PyTorch and experimented on a single Tesla V-100 GPU. We keep the batch size as one for a fair comparison. The experiment results show that CTPT is training efficiency compared to EmotionFlow and SPCL. As shown in Table 3, CTPT requires less training time a less GPU memory than EmotionFlow and SPCL while offering a better validation performance.

| | T5 | RoBERTa |
|---|---|---|
| TSPT | 0.4996 | 0.4884 |
| TSPT + CTPL | 0.5193 | 0.5110 |
| CTPT (w/o. BP) | 0.5571 | 0.5239 |

Table 5: Results of using different backbone models in DailyDialog.

**Analysis in Source Data** To explore the impact of external source data, we remove some external source tasks for MELD and IEMOCAP. For fair comparisons, we sample all the possible combinations of external source tasks [3] and report the average score. As shown in Figure 2, both MELD and IEMOCAP perform better when increasing the number of external source tasks. However, though different combinations bring different improvements, the average score improvement is likely linear.

**Analysis in Pipeline Component** In this section, we examine the influence of various components within our CTPT framework. Initially, we carried out ablation experiments to investigate the effect of integrating emotional knowledge into CTPL. As depicted in Table 4, the incorporation of emotional knowledge enhances the performance of the downstream task, highlighting its importance.

Further, to understand the contributions of CTPL and CTPO, we performed ablation experiments by omitting these components. Comparing the result of "TSPT + CTPL" with "CTPT" in Table 2, we can conclude that CTPO is important to CTPT since the validation performance will be significantly degraded without CTPO. Though the experiment result shows that CTPL is negative to the validation performance in some tasks like EC and IEMOCAP, adding CTPO will be positive. In summary, while CTPL's cross-task knowledge might not consistently enhance validation performance, due to the potential inclusion of redundant information, its combination with CTPO proves advantageous.

**Analysis in Different Backbones** As highlighted in Section 4.3, we selected T5, one of the

---

[3]For example, when the number of external source tasks is two, there are $C_4^2 = 6$ possible combinations.

| Source Task \ Target Task | EC | DailyDialog | MELD | EmoryNLP | IEMOCAP |
|---|---|---|---|---|---|
| EC | ╲ | **0.5119** | **0.2438** | 0.0307 | 0.1684 |
| DailyDialog | **0.5276** | ╲ | 0.2400 | 0.0308 | 0.2204 |
| MELD | 0.4579 | 0.4834 | ╲ | 0.0245 | 0.2313 |
| EmoryNLP | 0.3642 | 0.1804 | 0.1315 | ╲ | **0.2658** |
| IEMOCAP | 0.3870 | 0.2192 | 0.1104 | **0.0599** | ╲ |

Table 6: Performance of zero-shot transfers. The task-specific prompt of the target task is excluded during the training stage. We **bolded** the best zero-shot transfer result for each target task.

most emblematic text-to-text generative language models, as our primary backbone model. To further validate the generalizability of CTPT, we conducted additional experiments using an encoder-only backbone language model, RoBERTa.

As evidenced in Table 5, when employing RoBERTa as the backbone model, CTPT achieves results that surpass the TSPT baseline, comparable to those obtained with T5. This underscores the robust generalizability of our proposed CTPT across various backbone language models. It also suggests the potential for consistent enhancement in downstream task performance with the adoption of even more advanced backbone models.

### 5.3 Zero-Shot Transfer

In real-world scenarios, annotated training examples are not always available. Therefore, it is worthwhile exploring the zero-shot generalization ability of CTPT. In this subsection, we conduct experiments under the zero-shot setting that train the prompt by source task and evaluate the target task while excluding the external task-specific prompt from the target task.

As shown in Table 6, CTPT performs surprisingly under the zero-shot transfer. It outperforms baseline methods under the few-shot setting in EC, DailyDialog and IEMOCAP. Compared with the few-shot result of TSPT, CTPT zero-shot obtains better performance in DailyDialog and a sightly degradation in MELD and IEMOCAP.

Due to the similarity among the first three tasks (EC, DailyDialog, and MELD), the prompt trained by these three tasks can be easily transferred with each other, which almost achieves the result of TSPT under 16-shot. Meanwhile, since the conversations in EmoryNLP contain fine-grained and more complex emotions, prompts learned from other tasks can hardly be transferred to EmoryNLP. Specifically, the results of zero-shot transfer from the first three tasks to the rest two tasks are poor, and vice versa. In other words, the more similarity the two tasks have, the better zero-shot transfer performance they obtain. In summary, the experiment result shows that CTPT has a good generalization ability in zero-shot transfer.

## 6 Conclusion

In this paper, we strictly define the task of the few-shot setting for ERC and propose a cross-task prompt tuning (CTPT) method to tackle this problem utilizing the cross-task knowledge. CTPT learns from external task-specific knowledge from other tasks and emotional knowledge from commonsense and then summarizes the learned cross-task knowledge to improve the validation performance. We use a derivative-free optimization method to optimize the parameters without gradient information, which skips the backpropagation stage. Experiments on ERC benchmarks show that CTPT can outperform baseline models in the few-shot setting and obtain a surprising result in the zero-shot transfer. In summary, CTPT is training-efficient that includes: (1) *sample-efficiency*: it is trained by few-shot training examples, and (2) *computational-efficiency*: it tunes only about 1,000 parameters with derivative-free algorithms that skip the backpropagation.

## Acknowledgements

This research is supported, in part, by the Joint NTU-WeBank Research Centre on Fintech (Award No. NWJ-2020-007), Nanyang Technological University, Singapore. This research is also supported, in part, by the National Research Foundation, Prime Minister's Office, Singapore under its NRF Investigatorship Programme (NRFI Award No. NRF-NRFI05-2019-0002). Any opinions, findings and conclusions or recommendations expressed in this material are those of the authors and do not reflect the views of National Research Foundation, Singapore.

## Limitations

Though our proposed CTPT works well in source-limited scenarios, it has two main limitations:

- The DFO algorithm we use is under the sampling-and-updating structure so we need to compute the logits of all sampled candidate solutions to select the most optimal one. Meanwhile, the convergence speed of DFO algorithms is slower than backpropagation. Therefore, CTPT requires more forward passes than derivative-based methods due to the aforementioned limitations.

- In this paper, we use T5 as our backbone model. However, many large language models have been proven successful in other scenarios. It is worthwhile to explore how to utilize a larger language model under source-limited scenarios in future.

## Ethics Statement

In this paper, we do not involve extra ethical considerations:

- In this paper, we do not release any new data. All datasets we used are either public datasets or licensed for academic usage.

- In this paper, the source codes of baselines and other artefacts are open-sourced or licensed for academic usage.

- Our paper does not use demographic or identity characteristics information, and it does not harm anyone.

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
