# OpenReview forum: "Efficient Cross-Task Prompt Tuning for Few-Shot Conversational Emotion Recognition"
_EMNLP/2023/Conference — EMNLP 2023 Findings_

### Official Review · Reviewer_q8Be · 2023-08-03

**Soundness:** 4

**Excitement:**

4: Strong: This paper deepens the understanding of some phenomenon or lowers the barriers to an existing research direction.

**Paper Topic And Main Contributions:**

The manuscript proposes a derivative-free optimization approach CTPT to improve the sample and computational efficiency of ERC approaches. The main contributions are the derivative-free approaches, extensive experiment results, and convincing experimental results.

**Questions For The Authors:**

See Reasons to reject.

**Reasons To Accept:**

1. Derivative-free, and few parameters to optimize.
2. Extensive convincing results supporting the claimed contributions.

**Reasons To Reject:**

1. How significant is the function of emotional knowledge in the model? Is the impressive performance caused mainly by the model framework or this external knowledge?

2. Figure quality can be improved.

3. How sensitive is the optimization result to "A" and "p_0" in Equation (11)?



**Reproducibility:**

4: Could mostly reproduce the results, but there may be some variation because of sample variance or minor variations in their interpretation of the protocol or method.

**Reviewer Confidence:**

3: Pretty sure, but there's a chance I missed something. Although I have a good feel for this area in general, I did not carefully check the paper's details, e.g., the math, experimental design, or novelty.

---

> ### Author Rebuttal · Authors · 2023-08-29
>
> Thank you for your time and constructive feedback on our manuscript.
>
>
>
> 1. On the Significance of Emotional Knowledge versus the Proposed Cross-Task Prompt Tuning Framework (CTPT) in Performance Enhancement:
>
> Our proposed CTPT utilizes cross-task knowledge, which includes both External Task-Specific Knowledge and Emotional Knowledge. We have conducted an ablation study to analyse and compare the performance improvement induced by external task-specific knowledge and emotional knowledge. While both types of knowledge contribute to performance improvement, external task-specific knowledge shows a more substantial impact (see the following Table). These results suggest that emotional knowledge does offer additional performance gains. Nonetheless, even in the absence of emotional knowledge, the CTPT framework itself, using only external task-specific knowledge, significantly enhances performance, indicating its primary role in the observed improvements.
>
> |  Method  |  Performance on DailyDialog  |
> |  ----  | ----  |
> | TSPT  | 0.4996 |
> | CTPT w/o. Emotional Knowledge  | 0.5481 |
> | CTPT w. Emotional Knowledge  | 0.5571 |
>
>
> 2. On the Sensitivity of Optimization Results to Hyperparameters $A$ and $p_0$ in Equation (11):
>
> These two hyperparameters are randomly initialized and kept constant during the optimization. Our experiments with varying random seeds demonstrate that the optimization outcomes are not very sensitive to these parameters.
>
>
>
> 3. On Figure Quality:
>
> Thank you for pointing this out. We will enhance the clarity and quality of the figures in the revised manuscript.

---

### Official Review · Reviewer_ESEW · 2023-08-04

**Soundness:** 3

**Excitement:**

2: Mediocre: This paper makes marginal contributions (vs non-contemporaneous work), so I would rather not see it in the conference.

**Paper Topic And Main Contributions:**

This paper presents a derivative-free optimization method called Cross-Task Prompt Tuning (CTPT) for few-shot conversational emotion recognition. Unlike existing methods that learn independent knowledge from individual tasks, CTPT leverages sharable cross-task knowledge by exploiting external knowledge from other source tasks to improve learning performance under the few-shot setting. Moreover, CTPT only needs to optimize a vector under the low intrinsic dimensionality without gradient, which is highly training-efficient compared with existing approaches.

**Questions For The Authors:**

Question A: Why should we engage in few-shot learning for ERC when there is no lack of data and annotation is not particularly challenging?

Question B: ERC which solely focuses on cross-task applications seems to have limited significance as it can be viewed as a truncated form of data augmentation. Since datasets have inherent limitations while dialogue data is vast, wouldn't semi-supervised learning be more crucial in comparison to cross-task applications for data augmentation?

Question C: Since the performance of this approach is inferior to that of zero-shot chat GPT, what is the significance of this work?

==============================================================================================

I acknowledge the intriguing perspective of performing few-shot learning across tasks. However, if the achieved performance remains significantly lower than that of LLM in zero-shot scenarios, it becomes challenging for me to convince myself to embrace such an approach. For instance, observing a 30% performance gap on the MELD dataset and a 20% gap on the IEMOCAP dataset, it seems that the essence of few-shot learning might be compromised.
**Thus, I lean to keep my rating.**

**Reasons To Accept:**

- Reasonable method.
- The proposed method outperforms the baselines.

**Reasons To Reject:**

- The abstract looks a little unconvincing. For example, “ To improve both sample and computational efficiency, we propose a derivative-free optimization method called Cross-Task Prompt Tuning (CTPT) for few-shot conversational emotion recognition”. The conversational data is abundant, and few-shot learning may not be an appropriate choice. The author should clearly explain the rationale behind selecting few-shot learning in the paper and provide a more comprehensive justification for this choice.
- The motivation is rather ambiguous and lacks sufficient theoretical support. For example, “Existing prompt tuning methods independently learn task-specific knowledge from each task, yet such knowledge is often very limited in the few-shot setting ...”. It is recommended to clearly state the research objectives and significance in the introduction. Providing relevant literature and data to support the rationale behind the motivation is essential.
- Very difficult to follow the contribution of this paper. For example, “To the best of our knowledge, we are the first to strictly define and tackle the few-shot setting for the ERC task. We propose a Cross-Task Prompt Tuning (CTPT) method that can efficiently learn and utilize cross-task knowledge”. The author is not the first to propose few-shot learning in ERC. The author should provide a detailed explanation in the paper for its advantages compared to other methods.
- The significance of few-shot learning seems limited for the ERC task that has an abundant supply of data and labeling is not particularly challenging.
- Purely cross-task ERC has very limited utility, as it amounts to a truncated form of data augmentation. Dialogue data is infinite compared to the finite dataset, making semi-supervised approaches more crucial than cross-task augmentation. Additionally, it would be more meaningful to explore ERC research that spans different domains rather than just across tasks.
- Performance is inferior to that of zero-shot chatGPT.
- Table 2 and Table 4 need some significance tests to further verify the reliability of the experimental results..
- The quality of Figure 1 is not very clear. The author should incorporate vector graphics where appropriate to enhance the quality and readability of the paper.
- There is a lack of error analysis.

**Reproducibility:**

3: Could reproduce the results with some difficulty. The settings of parameters are underspecified or subjectively determined; the training/evaluation data are not widely available.

**Reviewer Confidence:**

4: Quite sure. I tried to check the important points carefully. It's unlikely, though conceivable, that I missed something that should affect my ratings.

---

> ### Author Rebuttal · Authors · 2023-08-29
>
> Thank you for your thoughtful review and important queries. We address your concerns as follows:
>
>
>
> ### Regarding Question A - The Motivation for Studying Few-Shot Setting in ERC:
>
> We acknowledge that from the standpoint of general NLP research, there is an abundance of well-labelled conversational datasets. However, the relevance of the few-shot setting should not be underestimated, particularly in the context of real-world applications. One illustrative example is the live chat customer service scenario explored in [1]. In such a scenario, it is highly critical to detect customer emotions. However, conversations with the same person often have very limited turns and are also highly domain-specific. The utterances are often very sparse in emotions, and most of the labels associated with utterances are neutral. In addition, many real application data are private and confidential, so it might be hard to collect large labelled conversation corpus for some highly domain-specific tasks. This challenge was also highlighted in [1]. Hence, we believe studying few-shot settings is an important step in translating the current ERC works into more real-world applications.
>
> Most existing studies concentrate on full dataset settings. [1] is among the first ones to investigate the few-shot setting. We go a step further by adopting a more stringent few-shot definition; specifically, we utilize only $k$ training examples per emotional category, whereas [1] employs substantially more than $k$. We compared our CTPT framework with existing ERC techniques under identical few-shot conditions, successfully demonstrating CTPT's effectiveness.
>
>
>
> ### Regarding Question B - Regarding the comparison with data augmentation and semi-supervised learning
>
> Different from data augmentation methods that usually generate some “virtual data”, our cross-task approach utilizes “real data” from other in-domain tasks. Hereby we do not need to consider the accuracy of the supervised signals. While semi-supervised learning may be a viable alternative, it still requires a large amount of unlabelled data for training. In contrast, our CTPT can deal with situations even when unlabelled data is limited.
>
>
>
> ### Regarding Question C - Comparison of the proposed CTPT and ChatGPT
>
> We acknowledge the impressive performance of newly released LLMs including ChatGPT and GTP-4. However, it's important to note that our framework operates orthogonally to the backbone LLMs. Our proposed CTPT is a prompt tuning framework that can be applied to various backbone LLMs, including the backbone of ChatGPT. We could not run experiments with ChatGPT backbone as it requires enormous computing capacity. However, we have conducted our experiments on different backbone models, not only T5 but also other language models like RoBERTa. Across these diverse models, our framework consistently yields performance gains relative to their respective baselines. Thus, we have a strong rationale to hypothesize that our framework could also be beneficial when applied to newer LLMs like ChatGPT.
>
> Additionally, the computational and financial burdens associated with tuning and deploying LLMs like ChatGPT underscore the critical importance of few-shot learning, further motivating our research in this direction.
>
>
> ### Others
>
> We will also revise to improve the figure quality and add significance tests and error analysis.
>
> ### Reference
>
> [1] Guibon et al. Few-Shot Emotion Recognition in Conversation with Sequential Prototypical Networks. In EMNLP 2021.

---

### Official Review · Reviewer_iGEX · 2023-08-05

**Soundness:** 4

**Excitement:**

3: Ambivalent: It has merits (e.g., it reports state-of-the-art results, the idea is nice), but there are key weaknesses (e.g., it describes incremental work), and it can significantly benefit from another round of revision. However, I won't object to accepting it if my co-reviewers champion it.

**Paper Topic And Main Contributions:**

This paper proposed few-shot conversational emotion recognition approach based on pre-trained language models.
1, it is training-efficient for its partial refine design.
2, it employs no BP in refinement.
3, it has enhanced transferring effcient.
In evaluation, the proposed system has achieved close recognition quality comparing with SOTA TSPT even without BP refinement.

**Reasons To Accept:**

The usage of emotion-task related specific prompt in emotion task is interesting, it requires much less training resources to turn the pre-trained models into a specific emotion recognition task, and the evaluation results have stated the effectiveness of the proposed approch.

**Reasons To Reject:**

The main contribution of the proposed paper is the usage of TSPT on a pretrained LM (T5 indeed) for emotion recognition task. The idea is interesting and result is good, however, the main optimization approach TSPT is not novel: it is already employed in different NLP approaches.

**Reproducibility:**

4: Could mostly reproduce the results, but there may be some variation because of sample variance or minor variations in their interpretation of the protocol or method.

**Reviewer Confidence:**

4: Quite sure. I tried to check the important points carefully. It's unlikely, though conceivable, that I missed something that should affect my ratings.

---

> ### Author Rebuttal · Authors · 2023-08-29
>
> Thank you for taking the time to read and review our work.
>
> We concur that TSPT has already been employed in different NLP approaches. Our main contributions lie in greatly improving the ERC performance in few-shot settings and under low-resource scenarios. We believe these contributions can be important in closing the gap between existing LLM models and real-world applications which are often faced with sample- and resource-limited scenarios.

---

### Meta-Review · Area_Chair_oUwY · 2023-09-07

**Recommendation:** 4

**Metareview:**

This paper introduces a few-shot conversational emotion recognition approach using PLMs. The idea is based on a partial refinement design that is training-efficient, avoids backpropagation in refinement, and enhances transferring efficiency. The proposed system achieves high recognition quality, comparable to the existing TSPT method. Additionally, the paper presents Cross-Task Prompt Tuning (CTPT) which is a derivative-free optimization technique for few-shot conversational emotion recognition. CTPT leverages cross-task knowledge, improving learning performance by incorporating external knowledge from other source tasks.


The majority of the reviewers engaged in post-rebuttal discussions and all agreed that the soundness of this work is mostly sufficient with an acceptable level of excitement and generally reproducible results.

---

### Decision · Program_Chairs · 2023-10-07

**Decision:**

Accept-Findings

**Comment:**

This paper introduces a few-shot conversational emotion recognition approach using PLMs. The idea is based on a partial refinement design that is training-efficient, avoids backpropagation in refinement, and enhances transferring efficiency. The proposed system achieves high recognition quality, comparable to the existing TSPT method. Additionally, the paper presents Cross-Task Prompt Tuning (CTPT) which is a derivative-free optimization technique for few-shot conversational emotion recognition. CTPT leverages cross-task knowledge, improving learning performance by incorporating external knowledge from other source tasks.


The majority of the reviewers engaged in post-rebuttal discussions and all agreed that the soundness of this work is mostly sufficient with an acceptable level of excitement and generally reproducible results.